# Call detail record aggregation methodology impacts infectious disease models informed by human mobility

**Hamish Gibbs**[1]*, **Anwar Musah**[1], **Omar Seidu**[2], **William Ampofo**[3], **Franklin Asiedu-Bekoe**[4], **Jonathan Gray**[5], **Wole A. Adewole**[5], **James Cheshire**[1], **Michael Marks**[6,7,8], **Rosalind M. Eggo**[9]

**1** Department of Geography, University College London, London, United Kingdom, **2** Ghana Statistical Service, Accra, Ghana, **3** Noguchi Memorial Institute for Medical Research, University of Ghana, Legon, Accra, Ghana, **4** Ghana Health Service, Ministry of Health, Accra, Ghana, **5** Flowminder Foundation, Stockholm, Sweden, **6** Department of Clinical Research, London School of Hygiene & Tropical Medicine, London, United Kingdom, **7** Hospital for Tropical Diseases, University College London Hospital, London, United Kingdom, **8** Division of Infection and Immunity, University College London, London, United Kingdom, **9** Department of Infectious Disease Epidemiology, London School of Hygiene & Tropical Medicine, London, United Kingdom

* Hamish.Gibbs.21@ucl.ac.uk

**Data Availability Statement:** CDR mobility data used in this study was provided by Vodafone Ghana in partnership with the Flowminder Foundation and Ghana Statistical Service. This data

## Abstract

This paper demonstrates how two different methods used to calculate population-level mobility from Call Detail Records (CDR) produce varying predictions of the spread of epidemics informed by these data. Our findings are based on one CDR dataset describing inter-district movement in Ghana in 2021, produced using two different aggregation methodologies. One methodology, "all pairs," is designed to retain long distance network connections while the other, "sequential" methodology is designed to accurately reflect the volume of travel between locations. We show how the choice of methodology feeds through models of human mobility to the predictions of a metapopulation SEIR model of disease transmission. We also show that this impact varies depending on the location of pathogen introduction and the transmissibility of infections. For central locations or highly transmissible diseases, we do not observe significant differences between aggregation methodologies on the predicted spread of disease. For less transmissible diseases or those introduced into remote locations, we find that the choice of aggregation methodology influences the speed of spatial spread as well as the size of the peak number of infections in individual districts. Our findings can help researchers and users of epidemiological models to understand how methodological choices at the level of model inputs may influence the results of models of infectious disease transmission, as well as the circumstances in which these choices do not alter model predictions.

## Author summary

Predicting the sub-national spread of infectious disease requires accurate measurements of inter-regional travel networks. Often, this information is derived from the patterns of

is available to researchers by application to the Flowminder Foundation (website: https://www.flowminder.org/, email: info@flowminder.org). Use of this data was approved by the LSHTM Research Committee (Ref: 22477) and the Noguchi Memorial Institute of Medical Research (Ref: 048/20-21). Population data was publicly available 2020 constrained population counts in 100m grid cells (not UN-adjusted), downloaded from the WorldPop project (DOI: 10.5258/SOTON/WP00682). Code and spatial boundaries used in this study are available under the MIT licence from: https://github.com/hamishgibbs/ghana_cdr_aggregation.

**Funding:** The following funding sources are acknowledged as providing funding for the named authors. EDCTP2 (RIA2020EF-2983-CSIGN: HG, RME, MM). HDR UK (MR/S003975/1: RME). This research was partly funded by the National Institute for Health Research (NIHR) using UK aid from the UK Government to support global health research. The views expressed in this publication are those of the author(s) and not necessarily those of the NIHR or the UK Department of Health and Social Care (NIHR200908: RME). ESRC UBEL Doctoral Training Partnership (HG). UK MRC (MC_PC_19065: RME). The funders had no role in study design, data collection and analysis, decision to publish, or preparation of the manuscript.

**Competing interests:** The authors have declared that no competing interests exist.

mobile device connections to the cellular network. This travel data is then used as an input to epidemiological models of infection transmission, defining the likelihood that disease is "exported" between regions. In this paper, we use one mobile device dataset collected in Ghana in 2021, aggregated according to two different methodologies which represent different aspects of inter-regional travel. We show how the choice of aggregation methodology leads to different predicted epidemics, and highlight the conditions under which models of infection transmission may be influenced by methodological choices in the aggregation of travel data used to parameterize these models. For example, we show how aggregation methodology changes predicted epidemics for less-transmissible infections and under certain models of human movement. We also highlight areas of relative stability, where aggregation choices do not alter predicted epidemics, such as cases where an infection is highly transmissible or is introduced into a central location.

## Introduction

The volume of travel between geographic locations is widely used as an input to epidemiological models of disease transmission. Mobility data provides an approximate representation of the travel of a population between subnational areas by recording the movement of a sample of individuals. One such form of mobility data is Call Detail Record (CDR) data which are collected from the customers of mobile network operators and record mobile device connections to the cellular network.

Metapopulation models [1] informed by CDR mobility data have been widely used to study the dynamics of infectious diseases including influenza [2,3], rubella [4], malaria [5,6] cholera [7] dengue fever [8] Ebola virus disease [9,10], HIV [11] and COVID-19 [12,13]. Transmission models informed by CDR mobility data are particularly useful in low and middle income countries where there has been a widespread adoption of mobile devices. These data can address a lack of prior knowledge about inter-regional patterns of travel and in turn, can build greater capacity for disease surveillance and prediction. Understanding what factors influence estimates of population mobility will allow for more accurate interpretation of the results of infectious disease models which rely on human mobility data. In this paper, we focus on factors introduced at the CDR aggregation stage, where individual records from mobile subscribers are aggregated to describe population-level mobility.

CDR data used in infectious disease research is typically produced as an aggregated, censored network describing the volume of travel between pairs of locations in a specified time period. The aggregation of CDR data transforms sensitive individual-level data into a description of population-level mobility, thereby reducing the risk of disclosing personally identifiable information. CDR aggregates can highlight different aspects of human mobility, from describing the volume of travel between sub-national districts (as addressed in this study) [14,15], to recurrent travel to/from a home district [16,17]. CDR data may be limited by the size of the customer segment from which it is collected, or by the interaction between individuals and a mobile device. To address sparsity in the sample of individuals or travel behaviours captured in CDR data, the data may be rebalanced to better match official sources of travel data [18]. Alternatively, CDR data may be used as the empirical input to models of human mobility which can estimate volumes of travel in sparse areas [19].

Previous research has demonstrated how estimates of infectious disease can be altered by the movement model chosen to represent population mobility, although these movement models are informed by the same input parameters [19,20]. In our research, we investigate the

impact of differences at the level of inputs to movement models, as well as the influence of the choice of movement model itself. We show how movement models are sensitive to empirical inputs and how this sensitivity leads to differing predictions of infection dynamics by subsequent epidemiological models.

Methodological choices about CDR data aggregation have important implications for the reliability of infectious disease models informed by human mobility. In our research, we focus on the extent to which methodological choices used when aggregating CDR data impact estimates of population mobility [14]. We show how, given identical CDR datasets, two common methodological choices during the aggregation procedure produce different representations of an empirical movement network. The first is the "all pairs" methodology which retains the long distance network connections while inflating the number of reported travellers as a consequence; while the second is the "sequential" methodology which was designed to accurately reflect the volume of travel between locations but does not include long distance connections. These methods were implemented during the COVID-19 pandemic to provide rapid indicators of changes in human movement because of their low computational complexity in transforming large individual CDR datasets into useful representations of population-level mobility.

We use CDR data collected by Vodafone Ghana and processed by the Flowminder Foundation using the open source FlowKit software [21] to investigate the impact of CDR aggregation methods on estimates of human mobility and subsequently, on the results of modelled infectious disease dynamics. This data is the result of a partnership between Ghana Statistical Service, Ghana Ministry of Health, Ghana Health Service, Vodafone Ghana, and the Flowminder Foundation [22,23].

## Results

### Aggregation of CDR data

We used Call Detail Record (CDR) data from Vodafone Ghana, a mobile network operator which collects CDRs from subscribers to calculate billing charges. The data used in this study records all transactions between a mobile device and the cellular network including calls, text (SMS) messages, and data usage (this is sometimes referred to as XDR data). The approximate location of a device can then be estimated using the location of the connected cell tower. "Movement" of devices is derived from the sequence of locations in which a mobile device connects to a cell tower. We use CDR data aggregated to the level of districts (Administrative Level 2).

We explore the impact of two different CDR aggregation methodologies: "all pairs" and "sequential" [24]. Given the movement of one device through the same sequence of districts, these methodologies produce a different movement network, one representing all connections between transit districts and one recording only sequential connections (Fig 1A and 1B). In the all-pairs network, long distance connections are retained at the expense of over-estimation of the volume of travel in the network, as devices may be counted more than once. Alternatively, in the sequential network, the number of trips in the network accurately represents the quantity of devices in the movement network while omitting long-distance connections (Fig 1C). Further, the sequential network will maintain a constant relationship between transit locations and network connections (Fig 1D) while in the all pairs network, an increase in the number of transit locations will accelerate the number of network connections for each device (and thereby increase the overall density of the network).

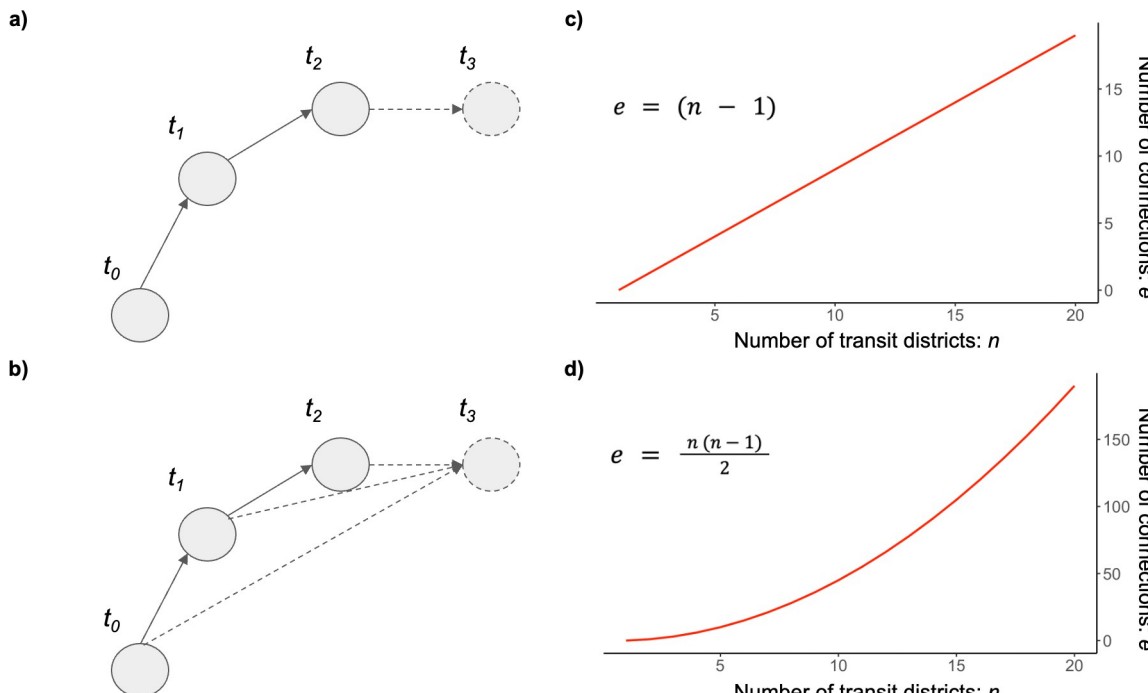

**Fig 1. Differences between CDR aggregation methodologies.** Synthetic networks showing the transport network created by a single device moving through four districts, aggregated using a) the sequential methodology and b) the all pairs methodology. The number of network connections with increasing numbers of transit districts in c) the sequential network, and d) the all pairs network.

## Differences in estimated population movement

We found that the all pairs methodology recorded an average of 2.33 million daily trips while the sequential methodology recorded 1.35 million trips (41% fewer trips) (Fig 2). For individual origin-destination pairs common to both networks, the sequential network had an average of 43% less travel compared to the all pairs network (Fig A in S1 Text). Aside from a higher overall volume of travel, the all pairs network was also more connected than the sequential network, with 13,523 connections compared to 5,805, a 57% difference. The all pairs network was also more dense (a comparison of the number of observed connections and the number of possible connections) compared to the sequential network (0.18 compared to 0.08 for the sequential network) (Table 1). The higher density of the all pairs network is likely a result of the increased number of trips and the uneven distribution of cell sites in Ghana (Figs B and C in S1 Text).

## Impact of aggregation on modelled human movement

Overall, each movement model reflected the differences in the empirical networks, with more connections and daily trips between districts in the all pairs methodology. However, the size of these differences varied based on the construction of the movement model (Fig 3 and Table 2). The power law gravity model produced a near-fully connected network based on both aggregation methodologies but a large difference in the number of modelled trips (+46% more trips in the all pairs network). The exponential gravity model produced a less connected network overall, with greater differences in the number of connections (+39%), but somewhat smaller differences in the number of modelled trips (+43%) in the all pairs network. The radiation model produced smaller differences in the number of trips between aggregation methodologies

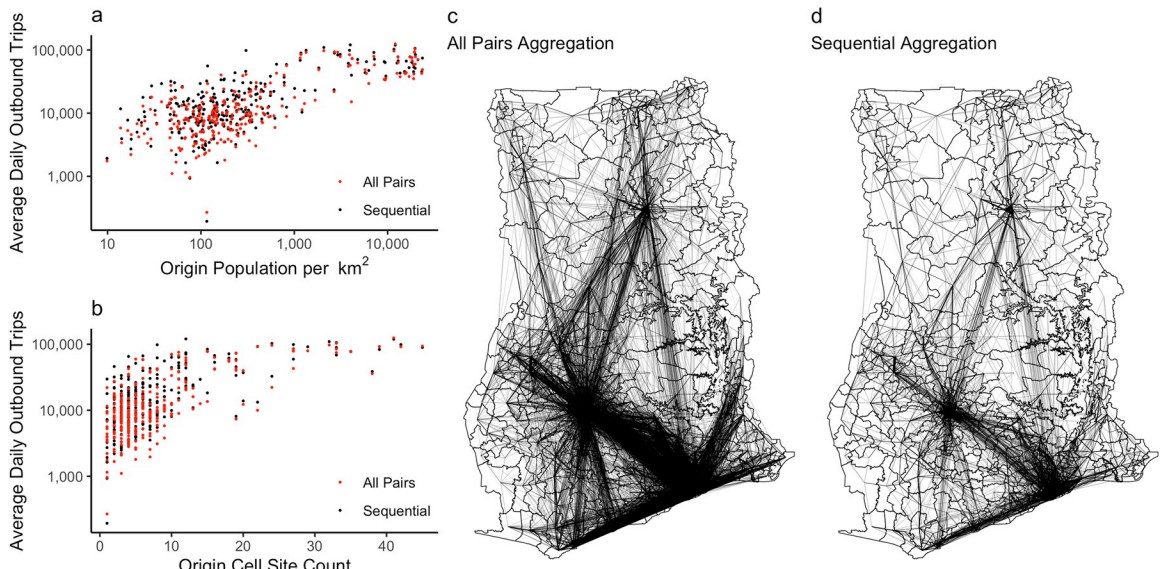

**Fig 2. Aggregation methodology increases reported movement with identical underlying CDR data.** The number of recorded trips relative to a) district population and b) the number of cell sites in a district. c) The all pairs movement network and d) the sequential movement network. Base map data are publicly available under the MIT licence from: https://github.com/hamishgibbs/ghana_cdr_aggregation.

(+38% in the all pairs network), but produced a less connected network compared to the power law gravity model or exponential gravity model (for sequential connections only).

We compared the modelled movement networks to the underlying empirical networks, finding that the radiation model had the lowest overall Mean Absolute Percentage Error (MAPE) and highest $R^2$ values compared to other models, indicating closer fit with the empirical data, but produced higher Root Mean Squared Error (RMSE) compared to the other models (Table 3). This likely indicates that the radiation model had better overall fit but introduced large errors for connections between certain locations. Because models were trained on different underlying empirical networks, these measures of model performance cannot be compared between different aggregation methodologies.

We compared the empirical networks and modelled networks, and calculated differences between aggregation methodologies for both the empirical and modelled networks predicted by each movement model (Figs 4, D, E, and F in S1 Text). Overall, we observe greater difference in the number of travellers recorded by different aggregation methodologies with respect to distance, as the length of connections increases, there is greater difference between the empirical networks. For the predictions of movement models, the difference between aggregation methodologies reflects the underlying construction of each model. This is especially evident in both gravity models (Figs 4B, 4C, Dd, and Ed in S1 Text), whereas the difference in the

**Table 1. Differences in observed movement caused by aggregation methodology.** Differences between two movement networks computed from the same underlying CDR data.

|  | All pairs | Sequential |
|---|---|---|
| *Total Connections* | 13,523 | 5,805 |
| *Daily Trips* | 2,331,125 | 1,354,908 |
| *Average Degree of a district* | 99.8 | 42.8 |
| *Network Density* | 0.18 | 0.08 |

 

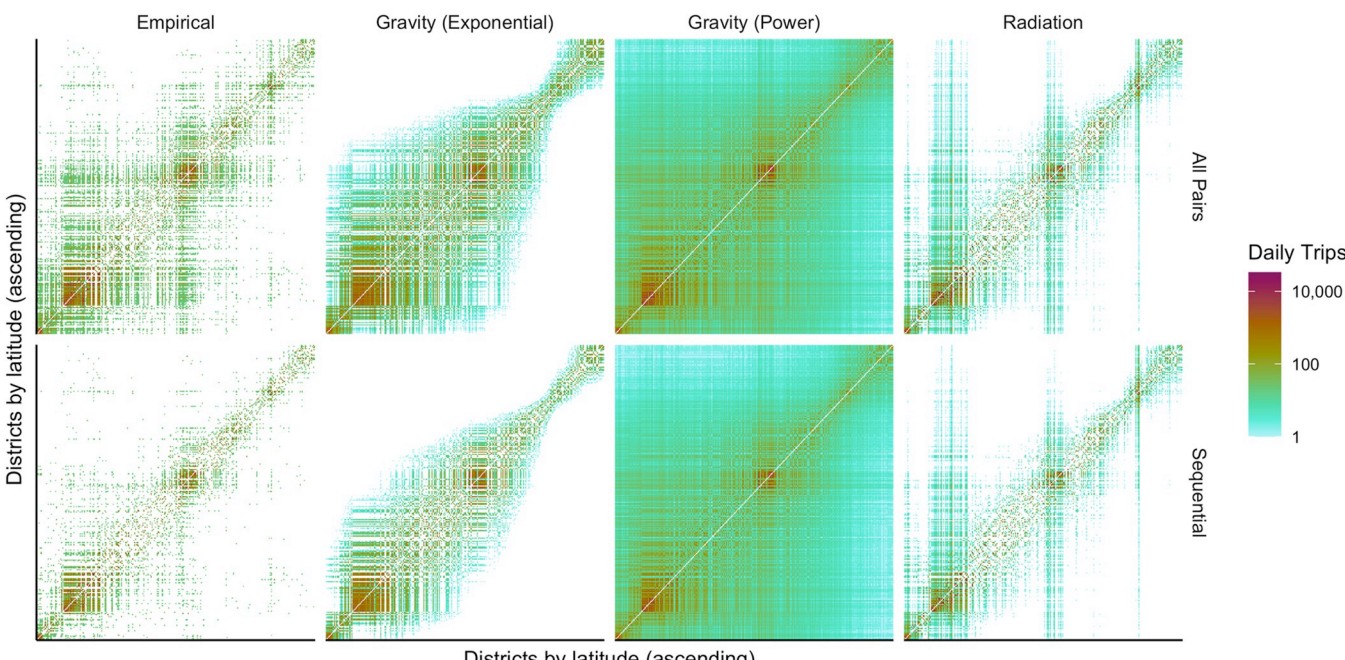

**Fig 3. Empirical and modelled networks informed by different aggregation methodologies.** Comparison of the empirical movement networks (left) and modelled networks for three types of movement models. Higher numbers of travellers in the all pairs network translates into a higher number of modelled travellers for all models.

radiation model (Figs 4D and Fd in S1 Text) more closely approximates the difference observed in the empirical networks.

## Results of aggregation methodology on an SEIR metapopulation model

We found that different aggregation methodologies influenced the epidemic trajectories of a stochastic SEIR metapopulation model but that this influence was highly sensitive to the choice of mobility model and $R_0$ (Fig 5). Under mobility models that produced a more sparse mobility networks (the Exponential Gravity and Radiation models), the use of the all pairs aggregation methodology resulted in significantly different infection dynamics including an earlier epidemic peak, higher peak number of infections, shorter epidemic length, and earlier average time of infection arrival in each district (Table 4 and Figs 5, G, and H in S1 Text). This finding, however, was not consistent for the Power Law Gravity Model, which produced very similar epidemics irrespective of the aggregation methodology. The differences in simulated epidemics were also influenced by the infectiousness of the epidemic, with aggregation methodology causing larger differences in epidemic characteristics for lower values of $R_0$.

**Table 2. Differences in travel network characteristics by movement model and aggregation methodology.** The difference in the number of modelled connections and daily trips using different models of human movement. The difference between empirical networks is reflected in predictions from each movement model.

| Model | All pairs aggregation | | Sequential aggregation | |
|---|---|---|---|---|
| | Connections | Daily Trips | Connections | Daily Trips |
| Gravity (Exponential) | 22,764 | 2,639,262 | 13,979 | 1,503,146 |
| Gravity (Power) | 73,170 | 3,958,835 | 73,168 | 2,137,533 |
| Radiation (Basic) | 16,886 | 2,148,336 | 14,184 | 1,332,434 |

**Table 3. Evaluation of movement models for different aggregation methodologies.** The Root Mean Squared Error (RMSE), Mean Average Percentage Error (MAPE), and $R^2$ comparing modelled movement to the empirical movement networks. Note that because models were informed by different empirical networks created from different aggregation methodologies, the evaluation cannot be compared between methodologies.

| | All pairs aggregation | | | Sequential aggregation | | |
|---|---|---|---|---|---|---|
| **Model** | **RMSE** | **MAPE** | **$R^2$** | **RMSE** | **MAPE** | **$R^2$** |
| Gravity (Exponential) | 764.44 | 1.72% | 0.33 | 659.74 | 2.33% | 0.24 |
| Gravity (Power) | 742.73 | 1.21% | 0.51 | 655.94 | 1.80% | 0.41 |
| Radiation (Basic) | 1,292.40 | 0.95% | 0.65 | 725.92 | 0.91% | 0.68 |

To understand whether differences in the progression of simulated epidemics were driven by differences in the topology of the transmission network, or merely indicated a "slowing" of similar infection trees under the sequential methodology, we calculated Spearman's rank correlation coefficient for the timing of epidemic arrival in each district. The Spearman rank correlation coefficient shows significant similarity in the sequences of infected districts under different aggregation methods, with a minimum correlation of 0.86 for all parameter combinations, with many simulated epidemics near or equal to 1 under either aggregation methodology (Table A in S1 Text). This indicates that, although epidemics may exhibit differences in the speed with which an infection spreads, these differences are largely driven by increases in the volume of travel around the movement network, not a change in the topology of the transmission network, which would result in different sequences of infected districts.

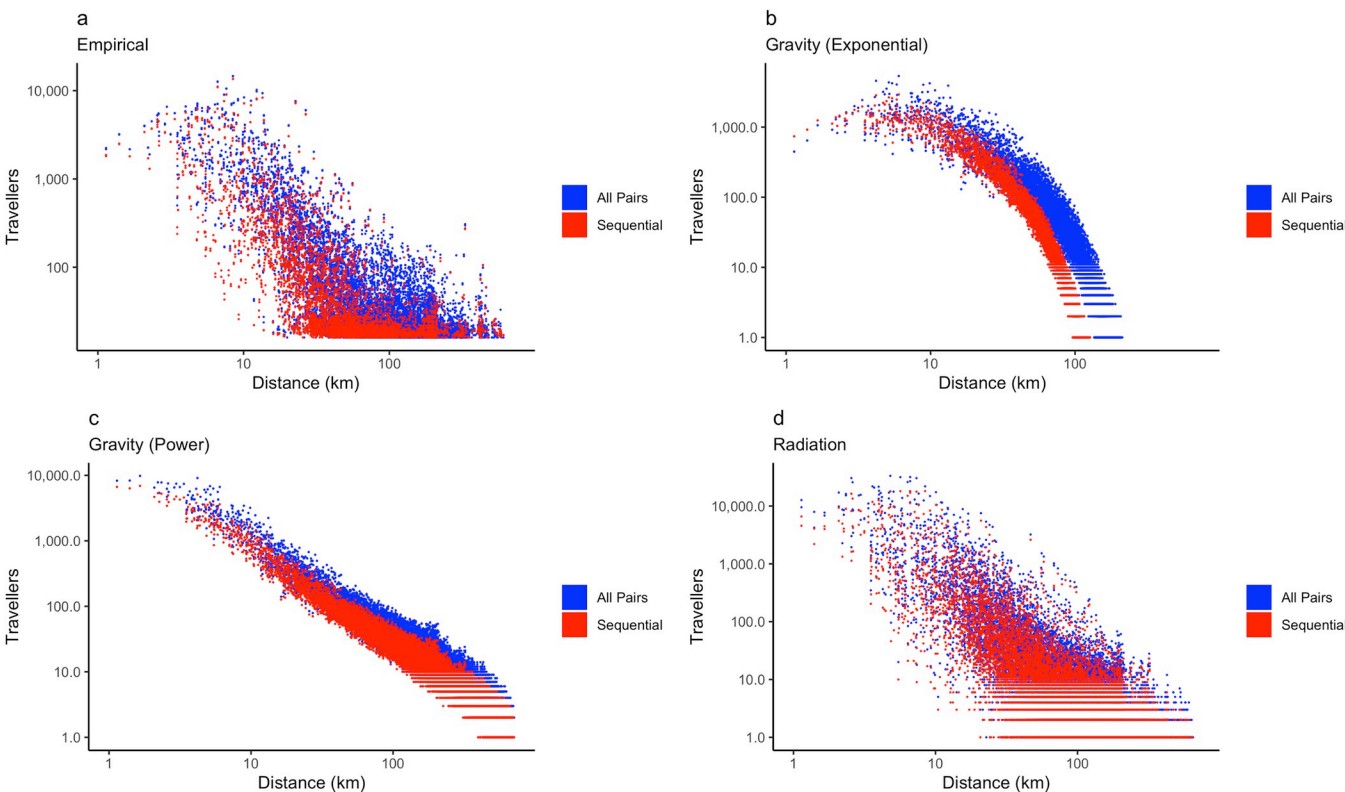

**Fig 4. Comparison of empirical and modelled travel networks by aggregation methodology.** The choice of aggregation methodology results in lower number of travellers in the sequential network in a) the empirical network, b) the exponential gravity model, c) the power law gravity model, and d) the radiation model, thereby leading to an underestimation of the number of travellers compared to the all pairs network.

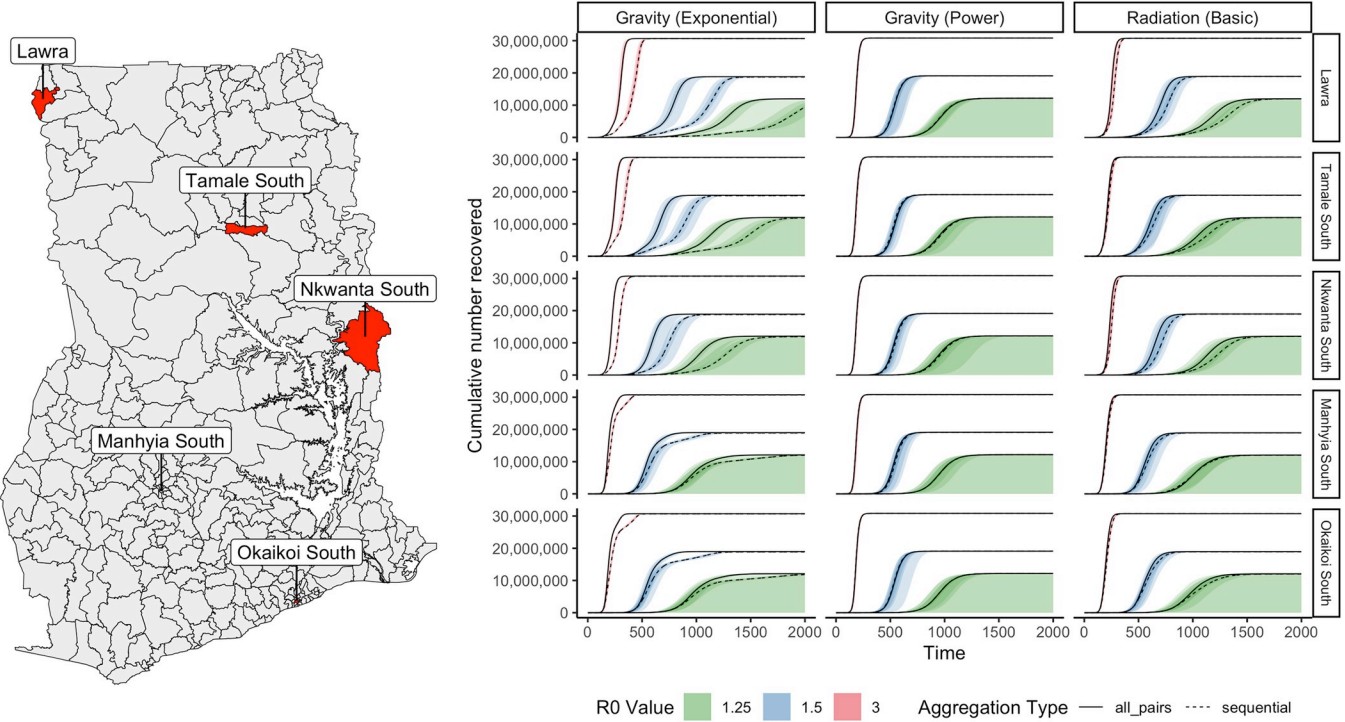

**Fig 5. Comparison of recovered individuals in simulated national epidemics by aggregation method, movement model, and $R_0$.** Density intervals show median, 20%, 60%, and 90% density distributions for 100 epidemic simulations for each parameter combination. Difference between solid and dotted lines indicates variation in the progression of national epidemics caused by aggregation methodology. Tamale South, Manhyia South and Okaikoi South are urban districts in the 3 largest cities of Ghana, and Nkwanta South and Lawra are rural districts. Base map data are publicly available under the MIT licence from: https://github.com/hamishgibbs/ghana_cdr_aggregation.

**Table 4. Difference in epidemic characteristics between aggregation methodologies.** The results of independent samples t-tests: 95% confidence intervals and p-values testing differences between the characteristics of simulated epidemics under different aggregation methodologies. Confidence intervals indicate the range of difference between the Sequential and All Pairs methodologies for each quantity (i.e. Column 1, Row 1—epidemics in the All Pairs network had an epidemic peak 96.91 to 149.04 days earlier compared to the sequential network). N/A values indicate some epidemic simulations that did not end in the modelled time period with R = 1.25.

| | $R_0$ | Gravity (Exponential) | Gravity (Power) | Radiation (Basic) |
|---|---|---|---|---|
| **Epidemic peak time (days)** | 1.25 | 96.91 to 149.04 (<0.0001) | -13.53 to 24.44 (0.5735) | 29 to 71.31 (<0.0001) |
| | 1.5 | 99.57 to 121.62 (<0.0001) | -3.03 to 6.59 (0.4688) | 27.69 to 39.89 (<0.0001) |
| | 3 | 49.83 to 57.82 (<0.0001) | 0.18 to 0.85 (0.0023) | 11.52 to 14.03 (<0.0001) |
| **Epidemic peak infections** | 1.25 | -94,957.63 to -79,615.91 (<0.0001) | -8,685.66 to 10,724.89 (0.8368) | -55,633.99 to -40,522.18 (<0.0001) |
| | 1.5 | -335,324.94 to -315,072.41 (<0.0001) | -4,070.94 to 19,049.5 (0.2041) | -158,879.79 to -141,001.94 (<0.0001) |
| | 3 | -1,683,016.56 to -1,643,336.03 (<0.0001) | -40,064.16 to -35,837.81 (<0.0001) | -560,367.84 to -512,214.82 (<0.0001) |
| **Epidemic Length** | 1.25 | N/A | N/A | N/A |
| | 1.5 | 284.1 to 314.54 (<0.0001) | -7.01 to 16.48 (0.429) | 45.68 to 70.16 (<0.0001) |
| | 3 | 119.74 to 124.07 (<0.0001) | -1.13 to 1.56 (0.7552) | 17.01 to 20.19 (<0.0001) |
| **Average time of 1st infection** | 1.25 | 201.17 to 227.42 (<0.0001) | 14.49 to 22.18 (<0.0001) | 61.65 to 79.54 (<0.0001) |
| | 1.5 | 144.96 to 157.82 (<0.0001) | 8.12 to 11.69 (<0.0001) | 37.39 to 44.22 (<0.0001) |
| | 3 | 56.72 to 61.33 (<0.0001) | 2.92 to 3.54 (<0.0001) | 12.66 to 14.48 (<0.0001) |
| **Average time of 5th infection** | 1.25 | 169.96 to 200.19 (<0.0001) | -1.73 to 14.89 (0.1205) | 51.72 to 74.56 (<0.0001) |
| | 1.5 | 133.09 to 146.15 (<0.0001) | 0.15 to 5.47 (0.0387) | 34.06 to 41.59 (<0.0001) |
| | 3 | 54.9 to 59.43 (<0.0001) | 1.57 to 2.23 (<0.0001) | 12.12 to 13.95 (<0.0001) |

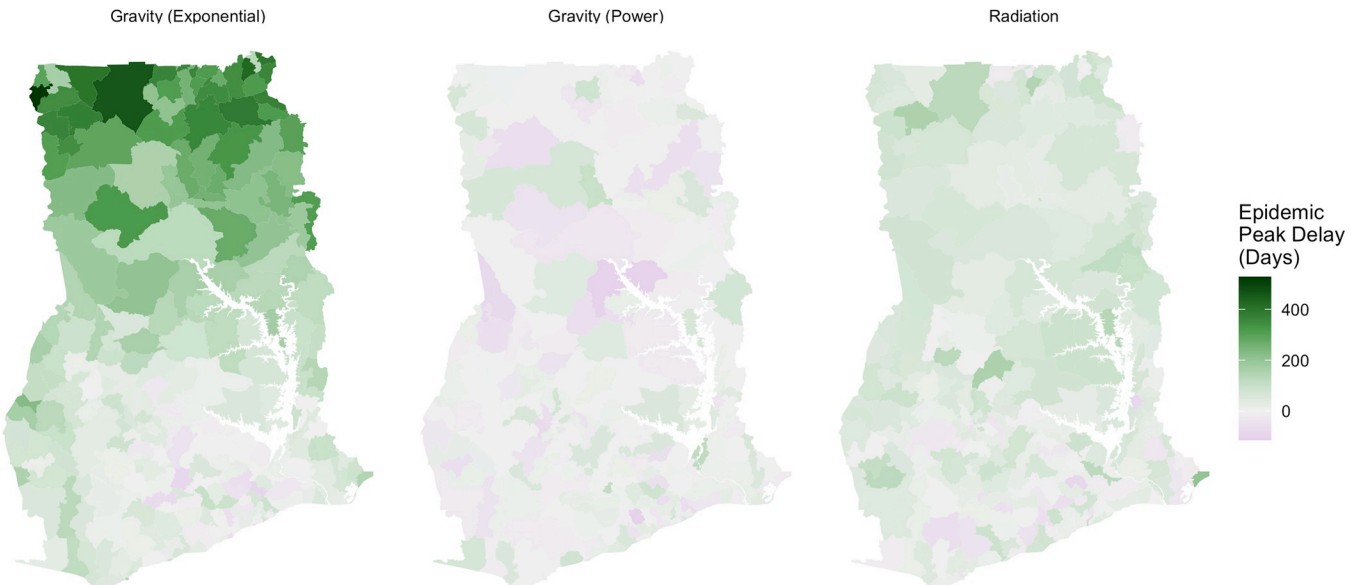

**Fig 6. Difference between the peak timings of national epidemics for different movement models.** Comparison of epidemic progression between aggregation methodologies for an epidemic with $R_0$ = 1.5 seeded in each district. Negative values indicate an earlier peak in the model informed by the all pairs network. Epidemics informed by different models of human movement show how differences in aggregation methodology vary spatially as a result of aggregation methodology and because of the choice of movement model. Base map data are publicly available under the MIT licence from: https://github.com/hamishgibbs/ghana_cdr_aggregation.

We found that the difference between aggregation methodologies was highly sensitive to the transmissibility and the location of infection introduction (Figs 6, I, and J in S1 Text). There was little difference between the timing and size of epidemics caused by aggregation methodology when an infection had higher $R_0$ or was introduced into central districts in the middle and southern parts of Ghana, which include the largest cities in Ghana: Kumasi and Accra respectively. However, the choice of aggregation methodology produced a greater difference in the progression of the modelled epidemic as infections were introduced into more rural locations, particularly in Northern areas, or were less transmissible ($R_0$ = 1.25, or $R_0$ = 1.5).

We found that the choice of movement model had a notable effect on the timing of national epidemic peaks between aggregation methodologies. This reflects the differences in the underlying construction of each movement model and their subsequent impacts on disease exportation between districts. The exponential gravity model, for example, produced a more sparse movement network than the power law gravity model, which led to a spatial regularity in the timing of the epidemic peak because the effect of aggregation has been exaggerated by the modelled travel networks. By contrast, the power law gravity model, which produced a more connected network overall, produced more spatial heterogeneity in difference between epidemic peaks, particularly for less transmissible infections ($R_0$ = 1.25) (Fig L in S1 Text). The arrival of infection based on the sequential aggregation methodology in certain districts may reflect the greater degree of randomness in the exportation of infections in a well-connected network. The radiation model showed less spatial heterogeneity, with delayed epidemics in the Northern areas of Ghana and around Lake Volta for less transmissible infections ($R_0$ = 1.25).

## Discussion

In this paper, we demonstrate the way that choices in the aggregation of CDR data can influence the results of models of human mobility and predictions of the spread of epidemics

informed by modelled human mobility. We show how two CDR aggregation methodologies used to respond to the COVID-19 pandemic produce different predicted epidemics in Ghana. The all pairs methodology, which produces a more densely connected network with higher volumes of travel, tends to produce an earlier arrival of the epidemic in all districts, with a higher peak number of infections. We show, however, that the difference between methodologies is sensitive to the transmissibility of infections, the location of infection introduction, and the choice of epidemic model and that in some cases, there is little difference between epidemics predicted using either aggregation methodology. For a highly transmissible infection ($R_0 = 3$), or an infection introduced into a fully connected travel network (such as the network produced by the power law gravity model), there is no significant difference between aggregation methodologies because infection is spread rapidly between locations, beginning local chains of transmission which dominate the dynamics of a predicted epidemic. However, for a less transmissible infection ($R_0 = 1.5$ or $1.25$), or for infection introduced into a more sparsely connected travel network, infections spread more gradually, especially when disease is introduced in a less central district, the choice of aggregation methodology produces significantly different epidemic progressions.

A large quantity of research has used CDR data as an input to transmission models for a range of infectious diseases in different national contexts. COVID-19 provided a new challenge for the use of CDR data because of the need to provide near-real time insights about population movement in a way that used limited computational resources to produce aggregated estimates of population movement. The aggregation methodologies considered in this paper were developed in this context, balancing the need to describe population movements with the need for rapid, low-complexity methods for their generation [17]. This need is particularly salient in low and middle income country contexts where computational resources are often constrained and there is limited capacity for disease surveillance, as well as in rapidly developing epidemics which require up to date estimates of changing patterns of travel behaviour.

The choice of CDR aggregation methodology is typically an initial step in modelling disease transmission, and as such, there is little research concerning the impact of these apparently minor methodological choices on the predictions of epidemiological models. However, there has been significant research interest in how different choices of mobility data, or different mobility models can impact the progression of an epidemic. This has included the comparison of empirical and modelled movement networks [19,25], varying mobility model constructions [20,26,27], and varying sources of empirical mobility data [2,16,28]. A key task for human mobility researchers is to understand areas of stability in epidemiological models informed by human mobility, where choices of mobility data or mobility model construction influence epidemiological models in predictable ways.

Previously published research has used a variety of methods to aggregate CDR data, from aggregating call volumes between pairs of locations, as addressed in this paper, to methods detecting changes in home locations defined by common presence [8] or night-time location [7,29]. Other research has used hybrid methods, such as recording travel to all administrative districts relative to device's home location [30]. Ultimately, the choice of aggregation methodology, as well as the spatial and temporal units used for aggregation, should be chosen based on a specific hypothesis in agreement with available understanding of underlying mechanisms of disease transmission. Future research can help to identify the circumstances, such as less transmissible infection or transmission in rural areas, under which methodological choices at the stage of CDR aggregation can produce variations in the predictions of epidemic models.

Because this is a novel study, we are unable to confirm the external validity of our findings across other contexts. We therefore call for further studies that use the same methodology to generate wider analysis of disease dynamics in sub-Saharan African countries other than

Ghana to assess whether the adoption of CDRs as inputs to epidemiological models will produce similar challenges as those found in the present study. However, we argue that our study is internally valid as extensive attempts were made to minimise any forms of systematic error that could potentially occur in this modelling exercise by accounting for variations in infection transmissibility, different models of human movement, and sensitivity to the location of infection introduction.

This paper focuses on the impact of methodological choices during the aggregation of CDR records. While these choices may result in important differences in observed patterns of movement, there are many other factors which influence the quantity and structure of movement captured by aggregated CDR data. These factors include uneven patterns of mobile phone usage and differences in individual travel behaviour. Some factors, like access to mobile devices or transportation may be further related to demographic characteristics like socio-economic status.

Technical factors may also alter the set of mobile devices included in CDR data. One such example is the uneven distribution of cell towers resulting in areas with minimal network connection. Mobile devices in these areas may be omitted from CDR data or may have a lower probability of generating CDRs regardless of the movement or activity of a given device. Cell towers are unevenly distributed in Ghana, clustering in population centres and along transportation networks. Cell tower density is also correlated with population, meaning that samples of CDR data may overrepresent devices located in more populous areas.

Despite the numerous factors which influence the movement behaviour represented by CDR data, we consider that the accelerating number of travellers relative to cell tower density observed in this study points to the influence of a considerably small methodological change on the level of movement in our dataset. Other factors do not influence the difference between the empirical networks or subsequent model outputs because both aggregates were produced from the same underlying CDRs. The influence of the aggregation methodology on observed levels of movement is further supported by the association between the observed volume of movement and the theoretical prediction of movement volume.

We have shown that aggregation methodology impacts the results of movement and epidemiological models informed by CDR data and that certain aspects of these models are more sensitive to the effect of CDR aggregation. While our findings should increase researchers' caution when using CDR aggregates, this source of mobility data remains invaluable for understanding patterns of human migration, particularly in low and middle income countries like Ghana. Moreover, CDR aggregates are widely used in operational settings, as the inputs to movement and epidemiological models and to inform government decision makers. The task of human movement researchers will be to continue to improve understanding of how these data can be used to reliably describe population movements, in spite of the shortcomings of CDR data.

## Materials and methods

### CDR mobility data

We used CDR mobility data collected by Vodafone Ghana and aggregated by the FlowMinder Foundation. This data was aggregated into districts (Administrative Level 2–271 districts). The boundaries of districts were defined by the government of Ghana. CDRs were assigned to an area based on the location of cell clusters within each region. A cell cluster is the location of a cell tower or the centroid location of a "cluster" of cell towers. In areas with a high density of cell towers, device connections may be "balanced" between multiple towers depending on network traffic and signal strength [31].

We used two aggregated versions of the same underlying CDR dataset collected between February 2021 and September 2021 which recorded the daily travel between a set of origin-destination pairs $(p_i, p_j)$ for each location $p$ within a set of locations $P$. The number of travellers between locations $w$ was defined as the total number of connections between pairs of locations. Pairs of locations with $w$ less than 15 were removed prior to data sharing to prevent identification of individual mobile devices. The matrix of OD pairs forms a weighted directed acyclic graph of travel between locations. We calculated the average number of travellers between pairs of locations across the data collection period for use in our analysis.

The CDR data used in this study do not include information on the district of residence for mobile devices, based for example, on where a device tends to be located at night. Instead, the data describe the daily number of travellers between pairs of districts. Data are recorded daily, and the aggregated mobility networks are the sum of travellers recorded across all individuals based on the sequence of districts to which a device connected each day.

## Population data

To define the population of administrative areas, we used 2020 population data from the WorldPop project [32]. WorldPop population data combines population counts from national censuses with remote sensing data using Random Forest-based dasymetric redistribution to estimate the population count across a surface of $100m^2$ cells. We used constrained population estimates, meaning that population counts match population counts from the Ghana Statistical Service, but were not adjusted to match UN national population estimates. We aggregated population estimates to administrative areas in Ghana using a spatial intersection of administrative boundaries with the population surface.

## Movement models

The empirical movement matrices used in this study included missing values where travel between pairs of locations did not exceed the censoring threshold of 15 trips during the study period. To fill in these missing connections, we used three common formulations of movement models to model missing connections in the empirical movement networks. This comparison allowed us to assess sensitivity of our findings to the choice of mobility model.

First, we used a power law gravity model defining the number of trips $\lambda_{i,j}$ between locations $i$ and $j$ as a function of the population size of the origin $N_i$, the destination $N_j$, and the distance between the origin and destination $d_{i,j}$ (Eq 1).

$$\lambda_{i,j} = \theta * \left( \frac{N_i^{\omega_1} N_j^{\omega_2}}{d_{i,j}^{\gamma}} \right) \tag{1}$$

In this model, travel between locations is defined by four parameters: a scaling parameter $\theta$, and weight parameters $\omega_1$, $\omega_2$, and $\gamma$, which alter the contributions of origin populations, destination populations, and distance respectively.

Second, we used an exponential gravity model defining the number of trips $\lambda_{i,j}$ between locations $i$ and $j$ using four parameters: a scaling parameter $\theta$, and weight parameters $\omega_1$, $\omega_2$, and $\delta$, which alter the contributions of origin populations, destination populations, and distance respectively (Eq 2).

$$\lambda_{i,j} = \theta * \left( \frac{N_i^{\omega_1} N_j^{\omega_2}}{e^{\frac{d_{i,j}}{\delta}}} \right) \tag{2}$$

Finally, we used a basic radiation model defining the number of trips $\lambda_{i,j}$ between locations

$i$ and $j$ as a function of the population size of the origin $N_i$, the destination $N_j$, the total number of trips leaving the origin $M_i$ and the population surrounding the origin $s_{i,j}$ defined by the population within the radius $r_{i,j}$ (Eq 3).

$$\lambda_{i,j} = M_i \frac{N_i N_j}{(N_i + s_{i,j})(N_i + N_j + s_{i,j})} \tag{3}$$

We fitted all mobility models using the Mobility [33] and rjags [34] R packages. Both gravity models were fitted to the empirical movement matrices using Markov Chain Monte Carlo (MCMC) parameter estimation. Both gravity models were fitted as likelihood functions with the number of trips specified as a Poisson distribution; whereas the weak informative prior distributions for the parameters $\theta$, $\omega_1$, and $\omega_2$, were defined by the Gamma distribution with shape and scale of 0.001 for parameter $\theta$ and with shape and scale of 1 for $\omega_1$, and $\omega_2$. The prior distribution of the parameter $\delta$ was modelled using a normal distribution truncated at 0 with mean and standard deviation calculated from the distance matrix. MCMC training was conducted using 4 chains of 50,000 samples each, with a burn-in of 10,000 samples. We assessed convergence using the $\hat{R}$ convergence diagnostic, requiring a threshold where all parameters were deemed valid with $\hat{R}$ less than 1.05.

We compared empirical and modelled networks by the total number of edges (connections) in the network, the total number of network trips (the sum of weights along each edge), the average node degree (the average number of edges connected to each node), and the network density (the ratio of the number of network edges compared to the number of possible edges). We also compared the performance of each model against the empirical data using Root Mean Squared Error (RMSE), Mean Absolute Percentage Error (MAPE) and $R^2$. We assessed the quality of model predictions by identifying the model with the lowest RMSE and MAPE, and highest $R^2$, indicating a close fit with the empirical travel network while minimising model error.

### Epidemiological modelling

We modelled the spread of infection using a stochastic metapopulation SEIR model implemented in the *R* package *SimInf* [35–37]. This model simulates an epidemic by modelling the transition of individuals in a connected sub-populations between compartments (Susceptible, Exposed, Infected, Removed). The model is stochastic, meaning that transitions between compartments are modelled through a random count measure and infection states for each subpopulation form a Continuous Time Markov Chain. The progression of the epidemic was modelled in individual subpopulations, defined by district boundaries, and connected by modelled movement networks. In the stochastic model construction, Infectious individuals are exported between subpopulations according to the average daily volume of movement between pairs of districts. For movement of an individual between subpopulations, individuals are sampled from a hypergeometric distribution with probabilities equal to the proportion of individuals in each compartment of the source population. Therefore, the probability that infections will be exported between subpopulations reflects the size of the epidemic within subpopulations as well as the volume of connections to other subpopulations.

Our model assumes constant rates of replacement (births) and mortality (deaths) within subpopulations during the study period. Although this assumption does not reflect real population characteristics, there is not sufficient data to estimate the rate of population change in Ghana during the study period. The model also assumes a uniform contact rate among members of a subpopulation. In reality, within-population contact rates vary relative to age structure and other demographic factors which are not captured by our model. The inclusion of

some age-structure adjustment within population-units, as opposed to population units only, would reduce the uncertainty and fine-tune the predictions from this analysis.

We assess the sensitivity of the model to the location of infection introduction by introducing 10 index infections into one district for a series of model simulations with different parameter combinations and introduction locations. Because of computational limits, we selected a random sample of 15 districts in Ghana and included an additional 5 districts: Nkwanta South (Greater-Accra Region), Manhyia South (Ashanti Region), Tamale South (Northern Region), and two rural districts: Lawra (Upper West Region) and Nkwanta South (Oti Region) to provide a representative sample of the rural and urban gradient of districts in Ghana. For these districts, we performed 100 epidemic simulations for three values of $R_0$: 1.25, 1.5, 3.0, and for each movement model: Gravity (Exponential), Gravity (Power Law), and Radiation (Basic). For all other districts, we performed 10 epidemic simulations for each parameter combination. We chose these values of $R_0$ to simulate an infection similar to COVID-19, with $R_0$ between 1 and 3. Because $R_0$ is not a model input parameter, we vary the transmission rate $\beta$, given a constant recovery rate, $\gamma$.

We assess the effect of aggregation methodology under different parameter conditions using two statistical tests. First, we perform independent sample t-tests comparing the epidemic quantities: epidemic peak timing, epidemic peak number of infections, epidemic length, and earlier average time of $1^{st}$ and $5^{th}$ infection arrival for simulations across all sampled districts (N = 2,000). These results indicate whether there is a significant difference between the average of each epidemic quantity between simulated epidemics informed by each aggregation methodology. Second, although aggregation methodology may change the characteristics of epidemics under certain conditions, similar epidemic characteristics could be driven by differences in the sequence of infection export between subpopulations, or could be caused by differences in the topology of the underlying infection network. To understand whether observed changes in epidemic characteristics are a result of increased "speed" of an epidemic, or are the result of changes in infection network topology, we calculate the Spearman rank correlation coefficient of the average arrival time of the $1^{st}$ and $5^{th}$ infections across 100 simulations. High correlation in the sequence of infections indicates greater similarity in the infection network, because of similarity in the sequence of infected districts between different aggregation methodologies.

## Supporting information

**S1 Text. Fig A in S1 Text. Comparison of individual origin-destination pairs between networks.** The difference in the average daily volume of travel for the 5,804 individual origin-destination pairs common to both networks. Note that because of the difference between aggregation methodologies, many origin-destination pairs are censored in the Sequential network. Further, note that some pairs have higher volumes of travel in the Sequential network compared to the All Pairs network. This is caused by empirical differences in the volume of travel on specific days (certain origin-destination pairs in the Sequential network may have high counts on particular days but otherwise are censored). **Fig B in S1 Text. The number of cell sites per district.** The spatial distribution of cell sites, showing a high density of cell sites in urban areas. Base map data are publicly available under the MIT licence from: https://github.com/hamishgibbs/ghana_cdr_aggregation. **Fig C in S1 Text. Number of cell sites by population.** The number of cell sites compared to the population in individual districts. **Fig D in S1 Text. Comparison of empirical and modelled travel networks.** a) Empirical networks from each aggregation methodology. b) Movement networks modelled using the exponential gravity model. Distance kernels show the number of travellers by the distance of network

connections in the c) empirical and d) modelled networks. **Fig E in S1 Text. Comparison of empirical and modelled travel networks.** a) Empirical networks from each aggregation methodology. b) Movement networks modelled using the power law gravity model. Distance kernels show the number of travellers by the distance of network connections in the c) empirical and d) modelled networks. **Fig F in S1 Text. Comparison of empirical and modelled travel networks.** a) Empirical networks from each aggregation methodology. b) Movement networks modelled using the radiation model. Distance kernels show the number of travellers by the distance of network connections in the c) empirical and d) modelled networks. **Fig G in S1 Text. Comparison of modelled national epidemics by aggregation methodology.** The difference in the number of individuals in the "recovered" compartment for a sample of 20 introduction locations, mobility models, and values of $R_0$. Epidemics were modelled 100 times for each combination of aggregation methodology, introduction location, $R_0$, and mobility model. **Fig H in S1 Text. Comparison of modelled national epidemics by aggregation methodology.** The difference in the number of individuals in the "infected" compartment for a sample of 20 introduction locations, mobility models, and values of $R_0$. Epidemics were modelled 100 times for each combination of aggregation methodology, introduction location, $R_0$, and mobility model. **Fig I in S1 Text. Influence of introduction location on the difference between aggregation methodologies.** Difference between the timing of the peak of a modelled epidemic with $R_0 = 3$. Negative numbers indicate that the predicted epidemic based on the all pairs methodology was later than the epidemic predicted based on the sequential methodology. Base map data are publicly available under the MIT licence from: https://github.com/hamishgibbs/ghana_cdr_aggregation. **Fig J in S1 Text. Influence of introduction location on the difference between aggregation methodologies.** Difference between the timing of the peak of a modelled epidemic with $R_0 = 1.25$. Negative numbers indicate that the predicted epidemic based on the all pairs methodology was later than the epidemic predicted based on the sequential methodology. Base map data are publicly available under the MIT licence from: https://github.com/hamishgibbs/ghana_cdr_aggregation. **Table A in S1 Text. Spearman correlation coefficient comparing sequence infection under each aggregation method.** Correlation approaching one indicates high similarity between the sequence of infected districts under each aggregation methodology. All correlation coefficients are significant with p-value <0.0001.
(DOCX)

## Author Contributions

**Conceptualization:** Hamish Gibbs, Anwar Musah, Rosalind M. Eggo.

**Data curation:** Hamish Gibbs, Jonathan Gray, Wole A. Adewole.

**Formal analysis:** Hamish Gibbs.

**Funding acquisition:** Michael Marks, Rosalind M. Eggo.

**Investigation:** Hamish Gibbs.

**Methodology:** Hamish Gibbs, Anwar Musah, Jonathan Gray, James Cheshire, Michael Marks, Rosalind M. Eggo.

**Software:** Hamish Gibbs.

**Supervision:** James Cheshire, Michael Marks, Rosalind M. Eggo.

**Validation:** Hamish Gibbs, James Cheshire, Michael Marks, Rosalind M. Eggo.

**Visualization:** Hamish Gibbs.

**Writing – original draft:** Hamish Gibbs.

**Writing – review & editing:** Hamish Gibbs, Anwar Musah, Omar Seidu, William Ampofo, Franklin Asiedu-Bekoe, Jonathan Gray, Wole A. Adewole, James Cheshire, Michael Marks, Rosalind M. Eggo.

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
