## [Decision Letter · Decision Letter 0]

14 Apr 2023

Dear Mr. Gibbs,

Thank you very much for submitting your manuscript "Call detail record aggregation methodology impacts infectious disease models informed by human mobility" for consideration at PLOS Computational Biology.

As with all papers reviewed by the journal, your manuscript was reviewed by members of the editorial board and by several independent reviewers. In light of the reviews (below this email), we would like to invite the resubmission of a significantly-revised version that takes into account the reviewers' comments.

We cannot make any decision about publication until we have seen the revised manuscript and your response to the reviewers' comments. Your revised manuscript is also likely to be sent to reviewers for further evaluation.

Sincerely,

Yamir Moreno

Academic Editor

PLOS Computational Biology

Virginia Pitzer

Section Editor

PLOS Computational Biology

Reviewer's Responses to Questions

**Comments to the Authors:**

Reviewer #1: The authors study the sensitivity of different aggregation methods of Call Detail Records (CDR), which can be used as a proxy for population mobility. They analyze differences in both the network structure and epidemic dynamics. Specifically, they compare two different aggregate methods: all-pairs and sequential. This paper has a very specific succinct goal, which the authors articulate well and succeed in reaching. As an epidemic modeler who could use these data/methods I think the paper could be strengthened with some additional analysis. I also have a couple questions about the initial conditions of the epidemic model that I did not fully understand from the methods section. Details below:

-The main epidemiological metrics discussed are the shapes of the epidemic trajectories and peak magnitude/timing. I would also be interested to know whether there are other epidemiological quantities that could be informative such as the length of the epidemic or attack rates.

-On the network comparison side of the work. I would be interested to see how much edge variation that is. Currently, the paper reports the macroscopic differences in the total trips and how the average daily outbound tips change with the population and cell site count. But I would be interested in the micro-level variation. For example for each source-target location pair how do the number of travellers vary. A simple visualization is a scatter plot of each pair where the x-axis

-When the difference in peak time is calculated is it only using 10 realizations? That doesn’t seem like enough statistics and I don’t think the process is described.

-I don’t think I understand how the initial conditions are set. In the methods sections it says that there are 100 index infections introduced to all districts in Ghana, but I thought the analysis only studies 5 source districts. Does 100 represent the initial number of infections or the number of realizations? The section then reports that 10 introductions are sampled, but does that refer the 10 epidemic curves shown in figure 4? The last paragraph in the method section is not very clear in describing the initial conditions.

-Minor question:Does the number of infections in the epidemic trajectory figure refer to the prevalence or incidence of infections in the SEIR model (daily?)?

Reviewer #2: In this study, the authors compare two different approaches to generate origin-destination matrices from mobile phone derived location data. Since individual trajectories from call-detail-records (CDR) can be aggregated into OD matrices in different ways, it is important to compare the results of different procedures. Moreover, as OD matrices are often used to parameterize spatial epidemic models, different aggregation methods could lead to different epidemic simulation outcomes. Indeed, this is what the study shows with numerical simulations.

Given the huge and ever-increasing interest in using mobile phone data to inform epidemic models, the study represents a useful addition to the literature, and it provides a good example of the impact of aggregation methodologies.

I believe the results will be of interest to the readership of PLOS Computational Biology.

On a less positive side, I have some suggestions that I hope will help improving the manuscript, before recommending it for acceptance.

1. I think the manuscript would greatly benefit from the addition of an initial subsection, at the beginning of the Results, that provides more details on the dataset used and the aggregation methodology. Given that the focus of the paper are the aggregation methods, I think the description in the Methods lacks some important details and that the aggregation part should be introduced earlier in the manuscript. More specifically:

a. The acronym CDR is used to refer to the data but then the authors explain that this includes calls, text messages and data usage. Usually, in the literature, CDRs refers to calls only and I would clarify this point early in the manuscript to avoid confusion.

b. It is not very clear what type of mobility is inferred from mobile phone data. Is it recurrent or not? Do the authors know the home location of a user? Can they distinguish between users? What do the OD matrices represent in this case? The number of daily movements made by different users between provinces? The number of distinct users moving between provinces?

c. To help the reader understanding these points, a schematic figure would be very helpful. I suggest adding a figure 1 that better explains the concepts of Table 1, and specifically describes how movements of one device are aggregated over time and in space.

2. Although the study is novel, there are several examples in the literature of similar studies where authors compared different mobility data sources, often mobile phone derived, and/or mobility models for epidemic purposes. The list would be long, I am only citing a few ones, but I think more references should be added to provide context.

For instance:

- Perrotta D, et al. (2022) Comparing sources of mobility for modelling the epidemic spread of Zika virus in Colombia. PLoS Negl Trop Dis 16(7): e0010565.

- Oidtman, R.J., et al. Trade-offs between individual and ensemble forecasts of an emerging infectious disease. Nat Commun 12, 5379 (2021).

- Kraemer, M.U.G. et al. Utilizing general human movement models to predict the spread of emerging infectious diseases in resource poor settings. Sci Rep 9, 5151 (2019).

- Engebretsen, Solveig, et al. Time-aggregated mobile phone mobility data are sufficient for modelling influenza spread: the case of Bangladesh. Journal of the Royal Society Interface 17.167 (2020): 20190809.

3. Similarly, there is some relevant literature about different methods to derive OD matrices from mobile phone data. This is a well-studied problem that deserves a more in-depth introduction for this study. First, it should be noted that “all-pairs” or “sequential” do not represent the only possible ways to derive OD matrices from mobile phone data. They are probably the most parsimonious ones, but more details could be used to derive commuting matrices. For instance, see:

- Bonnel, Patrick, Mariem Fekih, and Zbigniew Smoreda. "Origin-Destination estimation using mobile network probe data." Transportation Research Procedia 32 (2018): 69-81.

- Iqbal, Md Shahadat, et al. "Development of origin–destination matrices using mobile phone call data." Transportation Research Part C: Emerging Technologies 40 (2014): 63-74.

4. Finally, in the epidemiological analysis, it would be interesting to see, beyond the relative peak timing, whether the epidemic spreads between locations preserving the same order of infection or if this changes substantially. I am proposing something along the lines of what was done by Panigutti et al. (Royal Soc. Open Science 2017), by considering the ranking of locations by time of seeding (first infection recorded in a node). Although the epidemic may spread faster or slower on one network, the relative order of nodes that get infected may be preserved thus suggesting that the underlying network topology does not dramatically change.

**Have the authors made all data and (if applicable) computational code underlying the findings in their manuscript fully available?**

Reviewer #1: Yes

Reviewer #2: Yes

PLOS authors have the option to publish the peer review history of their article (what does this mean?). If published, this will include your full peer review and any attached files.

Reviewer #1: No

Reviewer #2: No
---

## [Decision Letter · Decision Letter 1]

17 Jul 2023

Dear Mr. Gibbs,

We are pleased to inform you that your manuscript 'Call detail record aggregation methodology impacts infectious disease models informed by human mobility' has been provisionally accepted for publication in PLOS Computational Biology.

Best regards,

Yamir Moreno

Academic Editor

PLOS Computational Biology

Virginia Pitzer

Section Editor

PLOS Computational Biology

Reviewer's Responses to Questions

**Comments to the Authors:**

Reviewer #1: The authors have commented on and addressed all of my concerns. I believe it is ready for publication.

Reviewer #2: I thank the authors for their extensive revision, which addressed my concerns in full. I recommend the manuscript to be accepted for publication.

**Have the authors made all data and (if applicable) computational code underlying the findings in their manuscript fully available?**

Reviewer #1: Yes

Reviewer #2: Yes

PLOS authors have the option to publish the peer review history of their article (what does this mean?). If published, this will include your full peer review and any attached files.

Reviewer #1: No

Reviewer #2: **Yes: **Michele Tizzoni

---

## [Editor Report · Acceptance letter]

1 Aug 2023

PCOMPBIOL-D-23-00115R1 

Call detail record aggregation methodology impacts infectious disease models informed by human mobility

Dear Dr Gibbs,

I am pleased to inform you that your manuscript has been formally accepted for publication in PLOS Computational Biology. Your manuscript is now with our production department and you will be notified of the publication date in due course.

With kind regards,

Anita Estes
